# Nutrients and Microbiota in Lung Diseases of Prematurity: The Placenta-Gut-Lung Triangle

**DOI:** 10.3390/nu12020469

**Published:** 2020-02-13

**Authors:** Fiammetta Piersigilli, Bénédicte Van Grambezen, Catheline Hocq, Olivier Danhaive

**Affiliations:** 1Division of Neonatology, St-Luc University Hospital, Catholic University of Louvain, Brussels 1200, Belgium; fiammetta.piersigilli@uclouvain.be (F.P.); Benedicte.vangrambezen@uclouvain.be (B.V.G.); Catheline.hocq@uclouvain.be (C.H.); 2Department of Pediatrics, Benioff Children’s Hospital, University of California San Francisco, San Francisco, CA 94158, USA

**Keywords:** microbiome, nutrients, prematurity, lung development, respiratory distress syndrome, bronchopulmonary dysplasia

## Abstract

Cardiorespiratory function is not only the foremost determinant of life after premature birth, but also a major factor of long-term outcomes. However, the path from placental disconnection to nutritional autonomy is enduring and challenging for the preterm infant and, at each step, will have profound influences on respiratory physiology and disease. Fluid and energy intake, specific nutrients such as amino-acids, lipids and vitamins, and their ways of administration —parenteral or enteral—have direct implications on lung tissue composition and cellular functions, thus affect lung development and homeostasis and contributing to acute and chronic respiratory disorders. In addition, metabolomic signatures have recently emerged as biomarkers of bronchopulmonary dysplasia and other neonatal diseases, suggesting a profound implication of specific metabolites such as amino-acids, acylcarnitine and fatty acids in lung injury and repair, inflammation and immune modulation. Recent advances have highlighted the profound influence of the microbiome on many short- and long-term outcomes in the preterm infant. Lung and intestinal microbiomes are deeply intricated, and nutrition plays a prominent role in their establishment and regulation. There is an emerging evidence that human milk prevents bronchopulmonary dysplasia in premature infants, potentially through microbiome composition and/or inflammation modulation. Restoring antibiotic therapy-mediated microbiome disruption is another potentially beneficial action of human milk, which can be in part emulated by pre- and probiotics and supplements. This review will explore the many facets of the gut-lung axis and its pathophysiology in acute and chronic respiratory disorders of the prematurely born infant, and explore established and innovative nutritional approaches for prevention and treatment.

## 1. Introduction

Acute and chronic lung disease of the premature infant have multiple origins spanning from fetal life to perinatal and postnatal exposures. Genomic variations have a profound influence on critical pathways involved in lung development, maturation and adaptation to various environmental challenges [1]. On the other hand, nutrients supply both the fuel and building blocks for these essential processes, and they have a profound influence on the homeostasis and responses of the developing lung. Deficiencies, excesses or imbalances of nutrients lead to developmental, functional and inflammatory lung disorders (Table 1).

The concept of the gut-lung axis has emerged as a cross talk between the two systems, mediated by the microbiome, with bilateral influences on health and disease. Initial studies have highlighted the role of the intestinal microbiome on the metabolic functions of various organs as well as immune responses. Only recently, the development of genomics and metagenomic approaches (including 16S ribosomal RNA profiling and shotgun-sequencing technology) has allowed a more detailed and extensive definition of organ-specific microbiomes. These technologies have permitted the discovery and exploration of a distinct microbiome of the airway and the lung, but its origin and timeframe around birth are still incompletely understood. The paradigm of a sterile fetal environment preserved by the amnion is now challenged by the emergence of intrauterine microbiota influencing fetal programming and development [24] and its relation to premature birth and neonatal respiratory diseases [25,26].

Premature birth causes an interruption of lung development and maturation, and at the same time exposes the neonate to profound alterations of these systems. Abrupt interruption of placental nutrition, lung exposure to a highly oxidant environment, disruption of the nascent microbiome by invasive procedures, hospital microbiota, broad-spectrum antibiotics and many other factors lead to acute and chronic lung disease. Bronchopulmonary dysplasia (BPD), a major cause of chronic lung disease and mortality in preterm infants, is characterized by impaired alveolar and vascular development determined by a combination of genetic factors [1], antenatal exposures and postnatal insults, with a central role played by inflammation. The goal of this review is to summarize the current concepts on nutrition, microbiomes and lung disease in the premature infant, and to delineate strategies and potential opportunities for prevention and treatment by acting on the nutritional and microbiotic balance.

## 2. Role of Antenatal and Postnatal Nutrition

### 2.1. Global Substrate Deficiency and Growth Failure

Global nutritional intake, both antenatal and postnatal, has a critical importance on lung development and the pathogenesis of respiratory disorders in the preterm infant. Intrauterine growth restriction (IUGR), a consequence of placental insufficiency and suboptimal transfer of oxygen and nutrients to the fetus, is associated with chronic lung disease of prematurity and altered lung function during infancy, which may last throughout adulthood [27]. In the infant born prematurely with an immature lung at risk of developmental disruption and highly vulnerable to exposures, IUGR is an additional contributor to acute/chronic lung disease. In a prospective cohort of preterm infants <30 weeks monitored during pregnancy, those with abnormal fetal doppler velocimetry and fetal growth failure had a 33% incidence of BPD compared to 7% in controls [28]. Postnatal growth failure is also a critical factor that contributes to chronic lung disease. Early alteration of body composition can be found starting in the first weeks of life for infants who will develop BPD compared to matched controls [29], which suggests ante- or perinatal metabolic reprogramming. In a large cohort study of 600 preterm infants 500–1000 g, a twofold higher incidence of BPD was observed between the first and fourth postnatal weight gain quartile, among other adverse outcomes [30]. Recommendations regarding global energy supply and amounts of specific macro- and micronutrients in healthy and sick preterm infants are multiple, sometimes controversial or contradictory, and abundantly discussed in the literature (e.g., [31,32]).

### 2.2. Carbohydrates

Glucose supply through the placenta directly influences lung maturation processes. High fetal glucose exposure due to maternal insulin-dependent diabetes mellitus (DM) increases the risk for respiratory distress syndrome (RDS) in late preterm and term infants [33]. This may be explained by altered glucocorticoid-mediated fetal lung development and delayed surfactant maturation due to insulin resistance in late gestation [34]. Conversely, suboptimal postnatal carbohydrate intake and delayed enteral nutrition predisposes to BPD in preterm infants [35].

### 2.3. Lipids

The effect of undernutrition on respiratory outcome is well established. In animals, 72 h fasting results in RDS, and induces a decrease of dipalmitoylsulfatidylcholine (DPPC) content in lung lavage fluid and an increase in minimal surface tension, indicating surfactant dysfunction [36]. Using the elegant methodology of simultaneous stable isotope tracers using plasma glucose, free palmitate and body water to quantify surfactant disaturated-phosphatidylcholine (DSPC) kinetics in-vivo, P. Cogo et al. showed the importance of dietary and endogenous fatty acids (FA) in surfactant homeostasis, and the role of kinetics alteration in the pathogenesis of BPD [37].

Whether excess FA intake in a high-fat diet and obesity in pregnancy (a condition increasingly prevalent in developed and developing countries) has any effect on neonatal respiratory function is still poorly established. In a large study on women with preterm premature rupture of membranes, maternal obesity was not associated with adverse neonatal outcome after adjustment for gestational age at birth [38]. Some animal data suggest that a maternal high-fat diet affects fetal lung development through an inflammation-induced placental insufficiency mechanism [39].

Specific fatty acid deficiencies, both antenatal and postnatal, may lead to respiratory disorders. Long-chain polyunsaturated fatty acids (LCPUFAs), in particular docosahexaenoic acid (DHA) and arachidonic acid (AA), are critical for organogenesis during fetal life and for retina and brain development, and are major regulators of inflammation prior to and following birth [10]. In an endotoxin animal model of acute respiratory distress syndrome (ARDS), a high-LCPUFA diet had detrimental effect on lung function, potentially by disrupting surfactant homeostasis [40]. In an animal hyperoxia-induced BPD model, both ante- and postnatal DHA supplementation decreased inflammation and improved alveolarization [41].

LCPUFAs are acquired during fetal life mostly through passive and carrier-mediated placental transfer, a process peaking during the third trimester of pregnancy. Premature birth disrupts this process and results in lower DHA and AA systemic levels and adipose tissue storage, rendering the preterm infant even more dependent on postnatal nutritional strategies. This congenital deficit can further deepen after birth in parenterally and enterally fed preterm infants. Parenteral lipid solutions are not tailored for the preterm infant’s needs and, since human milk LCPUFAs are highly variable, enteral feedings rarely reach full volume prior to the second or third weeks after delivery, with limited bioavailability due to impaired intestinal absorption and lipolysis [42]. In a retrospective human cohort study of <30 weeks premature infants, every 1% decline in DHA was associated with a 2.5-fold increased risk of BPD [8]. However, in the largest randomized controlled trial (RCT) to date including 1237 infants <29 weeks, postnatal DHA supplementation failed to prevent BPD, illustrating the complexity of the mechanisms involved [43]. A recent meta-analysis of 15 RCTs involving various LCPUFAs and mixtures lead to the same conclusion [9].

Intravenous lipid emulsions are composed of three elements: triglycerides, glycerol and phospholipids. Both saturated and monounsaturated fatty acids are synthesized from acetyl CoA derived from the metabolism of fat, carbohydrate or protein. Linoleic acid (C 18:2ω-6) and α-linolenic acid (C 18:3ω-3) are considered essential fatty acids from which most other LCPUFAs are metabolized. Total parenteral nutrition based on soybean lipid emulsion, which provides essential linoleic acid (LA) and gamma-linolenic acid, may not be adequate for preterm infants due to deficient downstream metabolic enzymes, and does not allow levels comparable to third trimester fetuses and term-born infants to be sustained [10]. However, 100% fish oil-based emulsions (Omegaven, Fresenius-Kabi, Bad Homburg, Germany) result in elevated DHA and EPA plasma levels, but negatively affect the level of AA, which is essential for brain and retina development as well as growth [13,44]. Newer-generation fish oil-based compound (SMOFLipid, Fresenius-Kabi) showed a benefit in decreasing BPD severity compared to an olive oil-based compound (ClinOleic, Baxter Healthcare SA, Wallisellen, Switzerland) in one small RCT [11], although another RCT failed to show any difference in BPD incidence [12]. A retrospective cohort study comparing SMOFLipid with an older, soybean oil-based product (Intralipid, Baxter Healthcare SA) showed no difference in BPD prevalence [44]. LCPUFAs also influence lung pathology through their effects on surfactant and inflammation (see the “inflammation and BPD” section). In conclusion, human data are too scarce to provide conclusive evidence favoring one specific intravenous lipid compound versus another one in regards to preterm lung disease at this stage.

### 2.4. Micronutrients

***Vitamin A***. Vitamin A (VitA) and, more specifically, its principal bioactive metabolite retinoic acid, is important in regulating early lung development and alveolar formation, contributing to branching morphogenesis, adequate formation and maintenance of the alveoli type II pneumocyte proliferation and synthesis of surfactant-associated proteins B and C through complex pathways modulating extracellular matrix proteins, most importantly elastin [27]. Recent animal data suggest that IUGR exerts its deleterious effects on lung development and alveolarization in part through disruption of the retinol pathway [45]. In extremely low-birth-weight humans (ELBW), postnatal intramuscular VitA supplementation reduces the incidence of BPD or death significantly, albeit slightly [2,46]. In a recent meta-analysis of four trials, VitA supplementation for ELBW showed significant benefits in oxygen dependency at the postmenstrual age of 36 weeks in survivors (841 infants, pooled risk ratio, 0.88) and length of hospital stay [4]. In a large RCT, antenatal oral VitA supplementation in vitamin-deficient mothers resulted in improved lung functional tests in offspring with 9–13 years follow-up compared to controls, indicating the longstanding beneficial potential of VitA for the fetus and newborn [3]. High pharmacy costs of intramuscular VitA preparations and the high number needed to treat have led certain neonatal intensive care units not to adopt this regimen; however, VitA prophylaxis may prevent BPD-related morbidity throughout infancy and childhood, and thus lead to numerous other long-term potential health benefits [47]. Disruption of the retinol pathway by genetic variants is a major contributor of congenital diaphragmatic hernia (CDH) [48]. Single-gene mutations or large genomic alterations may be involved in the non-isolated forms or subtle primary or secondary gene alterations in the isolated forms [49]. In a Japanese observational cohort, maternal dietary intake of VitA during pregnancy was inversely associated with congenital diaphragmatic hernia [50], suggesting that prenatal VitA supplementation may contribute to preventing this severe pulmonary disease.

***Vitamin D.*** The role of vitamin D (VitD) in non-skeletal health is increasingly recognized. Vitamin D plays relevant roles in placental development, in lung maturation processes and in the development of the innate and adaptive immune systems, and thus has the potential to influence neonatal lung pathophysiology through these various fundamental mechanisms. VitD receptor expression in the lung peaks in late gestation in animals, highlighting the role of vitamin D in alveolar maturation. VitD promotes AECII differentiation, enhances surfactant phospholipid and SP-B biosynthesis and stimulates surfactant release; in addition, VitD may be involved in postnatal lung growth and alveolarization, but evidence is limited (for review, see [51]). Preterm babies have lower levels of 25-OH-vitD at birth and are at a higher risk than term infants for postnatal deficiency due to intestinal immaturity and co-morbidities, which theoretically puts them at risk of acute and ch ronic lung disease [52]. However, the clinical impact of VitD on prematurity-related lung disease is not clear so far, with only a few studies linking RDS with VitD insufficiency (125-OH VitD 2–20 ng/mL) or deficiency (<12 ng/mL) [53,54]. Some epidemiological studies have associated neonatal hypovitaminosis D with risk of wheezing, asthma and respiratory tract infections later in infancy [55]. A prospective observational study correlating cord blood and 36-weeks corrected age 25OH-vitD levels in extremely preterm infants with odds of BPD failed to show any association, although most enrolled subjects did not reach vitamin D insufficiency levels (30/33 ng/mL at birth in the BPD and no-BPD groups, respectively) [56]. Interestingly, a similar study conducted in another environment with low/insufficient maternal and neonatal levels yielded opposite results (i.e., a significant correlation between maternal and neonatal levels between BPD/noBPD groups (19/28 and 15/7 ng/mL, respectively) [57]). One small RCT showed no difference in RDS or BPD in extremely preterm infants receiving low- or high-dose VitD supplementation [5]. Regarding the role of VitD on immunity, in one large RCT, 28–36 week preterm infants receiving high-dose VitD for six months had less wheezing than low-dose controls (OR 0.66) [6]. Altogether, these studies suggest that vitamin D insufficiency plays a role in BPD, but no clear evidence has emerged so far regarding the benefits and modalities of VitD supplementation for the purpose of neonatal lung disease prevention.

***Vitamin E.*** Vitamin E (VitE) influences fetal growth and early lung development [58] and is a potent antioxidant, but its relevance in preterm lung disease and the effects of supplementation are unclear to date. In a murine model of CDH and lung hypoplasia, maternal supplementation of α- and γ-tocopherol have led to increased lung complexity, accelerated alveolar growth and increased air surface [59]. In premature infants with respiratory distress syndrome, lower levels of VitE were associated with increased risk of developing BPD [60]. Unfortunately, trials of regular- and high-dose VitE supplementation in very low birth-weight infants showed variable results for BPD prevention and, potentially, an increased risk of sepsis and necrotizing enterocolitis. A Cochrane meta-analysis of these same studies reported no association between VitE and BPD, although wide confidence intervals suggest that these studies, even combined, are underpowered for drawing definitive conclusions [7]. However, none of these trials were conducted after 1991, and the nature and epidemiology of BPD has since profoundly evolved. No human trial has investigated antenatal supplementation and neonatal outcomes so far (for review, see [61]).

## 3. Lung Disease Mechanisms and Nutrients

### 3.1. Fetal Growth and Lung Development

In a large prospective cohort of very premature, very low-birth-weight neonates, intrauterine growth restriction increased the risk of death and BPD up to fourfold [62] through various mechanisms including impaired vascular and alveolar development [63] independently from RDS and acute lung disease severity [62]. These findings were confirmed, among others, in another large epidemiological study on extremely low gestational age neonates (the Extremely Low Gestational Age Newborn ELGAN study) in which the incidence of BPD was 74% vs. 49% in infants with *z* scores < −1 and ≥−1 respectively, with an odd ratio of 3.2, showing the primordial importance of IUGR on BPD pathogenesis [64]. The relative contribution of impaired substrate and oxygen delivery in IUGR is unclear, as well as their specific contribution in BPD pathogenesis. IUGR-related placental dysfunction translates into significantly decreased expression and activity of the amino-acid system A and system L transporters, of lipoprotein lipase and of lipoprotein receptors, but not of the glucose transporters GLUT1-4, which is either unaltered or increased [65]. In a mouse IUGR model combining altered diet and stress, alveolar simplification similar to BPD and decreased expression of the vascular endothelial growth factor (VEGF) pathway were observed, reflecting disrupted airway and vascular development [66].

### 3.2. Respiratory Distress Syndrome (RDS) and Surfactant Dysfunction

Surfactant is a complex mixture of lipids (90%) and four specific proteins (10%), surfactant proteins-A (SP-A), -B, -C and -D (10%). Of the surfactant lipids, 80–90% are phospholipids, the most abundant of which is the phosphatidylcholine (PC) class, the di-saturated dipalmitoylhosphatidylcholine (DPPC) and other phospholipids with saturated fatty acids in the one and two position of the glycerol moiety being the most critical for surfactant tension-active properties [67]. Glycerol is the main substrate for PC synthesis in the early neonatal period, when its concentration in circulation increases markedly, whereas later in life, glucose is the major source [68]. The fatty acids of surfactant phospholipids are synthesized in type II cells, taken up from the blood or derived from recycling.

Primary surfactant deficiency due to alveolar epithelium immaturity is the cause of respiratory distress syndrome of prematurity (RDS). In adults and children, qualitative or quantitative surfactant dysfunction plays a major role in acute respiratory distress syndrome (ARDS), affecting phospholipids and surfactant-specific proteins associated with extensive lung tissue inflammation. The same secondary surfactant inactivation also exists in several post-natal conditions both in term and preterm infants, such as meconium aspiration syndrome, sepsis, pulmonary hemorrhage and others [69]. A delayed-onset surfactant dysfunction associated with transient SP-B deficiency has been observed in preterm infants with severe RDS protracted beyond the first week of life [70], highlighting the importance of postnatal surfactant disruption in the pathogenesis of BPD.

IUGR also leads to surfactant dysfunction at birth. In an experimental placental reduction model (global oxygen/substrate depletion), the expression of SP-A, -B, and -C protein and mRNA was reduced in growth-restricted fetal animals compared to controls at ages corresponding to severe and late prematurity in humans [71]. Interestingly, IUGR induction by maternal hypoxia yielded divergent effects, increasing the levels of many hypoxia-inducible genes in the fetus including SP-B and -D as well as ABCA3, a lamellar body phospholipid carrier, and aquaporin-4, which is in part responsible for the airway absorption of fetal lung fluid at birth [34]. Maternal undernutrition during late gestation directly affects surfactant lipid levels in the immediate postnatal period and alters lung structural development [72] as well as decreasing the surfactant pool size [73]. Together, these results illustrate a specific role of fetal nutritional deficiency in IUGR-related respiratory complications in humans.

Maternal diabetes mellitus and excessive fetal carbohydrate exposure alters surfactant synthesis in term and near-term infants. Animals born from induced-DM mothers show impaired lung development and maturation, with decreased expression of surfactant proteins B and C and their regulatory factor FOXA2 associated with inducible nitric oxide synthase induction and generation of reactive oxygen species [74], a possible mechanism explaining respiratory failure in infants of diabetic mothers.

The composition of dietary fats plays a role in RDS and ARDS. In a model of endotoxin-triggered ARDS, adult animals fed with an unsaturated fatty acid diet (either LA or fish-oil) showed more severe vascular congestion, intra-alveolar edema and alveolar septa thickening than those fed with a saturated fatty acid diet (palmitate), suggesting that different substrates may influence lung pathophysiology through different mechanisms [40]. In animal models of premature lung disease, maternal DHA supplementation improved lung growth and enhanced fetal surfactant composition and synthesis [75].

The specific protein fraction of surfactant is also dependent on dietary substrates. Maternal protein intake may directly affect surfactant protein synthesis. Surfactant protein-A (SP-A) levels in the fetal lungs and amniotic fluid from protein-malnourished pregnant rats are lower than in those with normal protein intake [76]. Prenatal and postnatal serum SP-D concentrations are higher in IUGR newborns than in controls, inversely correlating with birthweight percentiles [77] and potentially reflecting the well-established intrauterine hypercortisolemia characterizing the IUGR state rather than specific protein intake deficiency.

Inositol, a key component of membrane phospholipids originating from diet or endogenous synthesis, is contained in high concentration in colostrum and breast milk. In a 1992 RCT, parenteral administration of inositol to premature infants 26–32 weeks with RDS significantly increased the odds of survival without BPD (71% vs. 55%) [14]. However, these results were not confirmed in subsequent studies, and a recent Cochrane meta-analysis encompassing 1177 infants concluded in not recommending its use [15].

### 3.3. Pulmonary Vascular Disease

Pulmonary arterial hypertension is a major factor contributing to morbidity and mortality in BPD [78] and other forms of neonatal chronic lung disease such as CDH [79]. Recent animal studies show that postnatal malnutrition associated with hyperoxia induces right ventricle and pulmonary arterial remodeling in growth-restricted pups [80]. Interestingly, the same authors showed that intestinal dysbiosis is associated with postnatal malnutrition in this model, and that probiotic supplementation prevents the onset of pulmonary hypertension in the growth-restricted group [81]. In a mouse model of hyperoxia-induced BPD, maternal DHA supplementation resulted in reduced inflammation and decreased pulmonary vascular disease [82] in the offspring. In a retrospective cohort of 138 infants born <28 weeks, the birth-weight-for-gestational age ration was a strong independent predictor of pulmonary hypertension in those with moderate-to-severe BPD, with an odd ratio of 4.2 in the <25th percentile group [83], highlighting the importance of antenatal substrate supply in the fetal onset of pulmonary vascular disease.

### 3.4. Inflammation and BPD

Bronchopulmonary dysplasia (BPD), the main cause of chronic lung disease and respiratory morbidity and morbidity in preterm infants, is characterized by impaired alveolar and vascular development determined by a combination of genetic factors [1], antenatal exposures and postnatal insults to the developing lung, with a central role played by inflammation (Figure 1). BPD most strongly correlates with lower gestational age [84], and low maternal DHA levels are associated with premature birth [85]. In a simplified view, ω-6 FA are pro-inflammatory whereas ω-3 FA are anti-inflammatory. ω-3 FA, found in fish oil, in particular DHA, are capable of partly inhibiting many aspects of inflammation including leucocyte chemotaxis, adhesion molecule expression and leucocyte–endothelial adhesive interactions, production of eicosanoids like prostaglandins and leukotrienes from the ω-6 FA arachidonic acid and production of pro-inflammatory cytokines. Enriching diets with DHA can change the inflammatory balance, modulating cell function and suppressing inflammatory responses [86]. In a retrospective observational study in preterm infants <30 weeks of gestation, decreased DHA levels in the first weeks of life were associated with an increased risk of BPD [8]. In a multicenter prospective randomized control trial (RCT) aimed at comparing neurological outcomes in maternal breast milk-fed preterm infants <33 weeks of gestation whose mothers had received either a high- or standard-DHA diet (the DHA for the Improvement of Neurodevelopmental Outcome in preterm infants DINO trial [87]), a lower incidence of BPD was observed in the high-DHA diet group for some subsets of cases (males and for <1250 g infants) [88]. These preliminary human data led to an adequately targeted and powered RCT involving over 1200 infants <29 weeks (the N3RO trial), which, surprisingly, showed no benefit of postnatal DHA supplementation for BPD prevention [43]. These contrasting data illustrate the complexity of the interactions between lipid nutrients and the developing lung, supporting the need for further translational research and clinical trials.

Lactoferrin, an iron-binding glycoprotein secreted by many epithelial cells and contained at high concentration in breastmilk (~1 mg/mL) with various antimicrobial properties, is a key component of innate immunity together with defensins, IgA and other peptides [89]. Supplementation with bovine lactoferrin, which is 70% analogue to human, significantly decreased late-onset sepsis and necrotizing enterocolitis but not BPD in preterm infants [90]. Six subsequent trials, meta-analyzed in a 2017 Cochrane review, confirmed these findings acknowledging low-quality results and the need for better data [16]. Subsequently, a recent multicenter, adequately powered RCT including 2203 infants <34 weeks (the Enteral Lactoferrin supplementation For very preterm INfants ELFIN trial) showed no difference in any of the measured morbidities, including necrotizing enterocolitis (NEC), late onset sepsis and BPD [17], dismissing lactoferrin for BPD prevention.

## 4. Microbiomics of the Lungs—The Gut-Lung Axis in BPD

All the microorganisms inhabiting the body (both symbiotic and pathogenic) are defined as microbiota, whereas the whole genome of all the microbial communities that colonize the body are referred to as the microbiome. The number of microbial cells is 10 times more the number of human cells. Furthermore, the number of bacterial genes is more than 100 times that of human genes. Microbiota have a pivotal role in the development of the immune system and the metabolic homeostasis of their host.

Most of the microorganisms that inhabit the human body do not grow in vitro, and therefore the whole microbiome could be thoroughly explored only when high-throughput genomic sequencing technologies became available. The Human Microbiome Project [91] was launched in 2008 by the National Institute of Health with the goal of characterizing the entire bacterial community colonizing the human body. This would allow the determination of whether there was an association between microbiome alterations and the onset of specific diseases.

In neonatology, most microbiome studies have focused on the relation of necrotizing enterocolitis with a particular microbiome [92,93]. It is now clear that there is a strict association between the type of microbial community of the gut and the risk of developing NEC. Indeed, in order to prevent the development of NEC it is now recommended to provide probiotics as soon as possible, so to restore a beneficial commensal flora [22]. Contrarily, the lower respiratory tract of healthy individuals was considered sterile in the past. Nevertheless, the presence of a resident microbial community in healthy lungs recently emerged, and research currently focuses on correlating particular lung microbiome profiles with disease.

Studies exploring the neonatal lung microbiome have mostly used bronchoalveolar lavage fluid samples or lung biopsies acquired via surgical sterile explants. In older children and adults, the lung microbiota has been defined in various pulmonary diseases, such as cystic fibrosis, asthma or chronic obstructive pulmonary disease [94]. In the preterm infant, a specific lung microbiome is present from the first days of life, and is dominated by *staphylococcus* and *ureaplasma* species [26]. At first clinicians considered fetal lungs to be sterile, and the acquisition of a microbial community in the lungs was attributed to a colonization occurring in the immediate post-delivery period, depending on the mode of delivery. Intuitively, vaginally delivered infants would acquire bacterial communities resembling the microbial vaginal environment (*Lactobacillus, Prevotella*), whereas C-section infants would acquire mostly skin microbiota (*Staphylococcus, Corynebacterium, Propionibacterium*) [95]. Actually, the immature lung’s microbiome origin is controversial. Several studies have shown that airway bacterial colonization is already detectable at birth, suggesting an antenatal origin [25]. When bacterial DNA sequencing techniques became widely available, the detection of microbiota in the placenta, amniotic fluid and fetus became possible both in mice and human [96,97,98], challenging the concept of the sterile womb. One human study reported the presence of DNA from the common intestinal bacteria *Lactobacillus* and *Bifidobacterium* in placental biopsies collected at term after an elective C-section without rupture of membranes or chorioamnionitis [99]. Very recently, combining metagenomic shotgun sequencing and targeted 16S techniques, Al Alam et al. demonstrated the presence of diverse and overlapping taxa in placenta, lung and intestine from 31 human fetuses harvested from 10 to 18 weeks of gestation [100]. However, a recent study based on placenta samples from two large cohorts of complicated and uncomplicated pregnancies showed that most bacterial signatures were acquired during labor and delivery or resulted from technical contamination, with the exception of the pathogen *Streptococcus agalactiae*, which was observed in 5% of pre-labor placentas [101,102], as echoed by others [103].

After birth, maternal breastfeeding influences the colonization and maturation of an infant’s intestinal microbiome, depending on early or late lactation stage, gestational age, maternal health and mode of delivery (Figure 1). The same *Bifidobacterium* and *Lactobacillus* strains identified in an infant’s gut microbiome are also found in mother’s milk, indicating that breastfeeding is a postnatal route of mother–child microbial exchange. However, the origin of these microbes, the complex dynamics of their transmission and their site specificities remain to be determined [104].

Can lung microbiota influence lung diseases? Inflammation can predispose to BPD, and lower respiratory tract infections are a recognized inflammatory trigger [105]. Thus, it is foreseeable that certain bacterial infections can predispose to BPD. For example, *Ureaplasma urealyticum* infections have been associated with the development of BPD [26,106]. Indeed, several small RCTs of postnatal prophylactic macrolide therapy in preterm infants for BPD prevention have been conducted, showing some efficacy in a meta-analysis [107].

Is the susceptibility to developing BPD only associated with pathogenic bacteria, or also with dysbiosis? In fact, it is been demonstrated that neonates with a reduced diversity of the lung microbiome have an increased risk of developing BPD [108]. In their systematic review, Pammi et al. evaluated six studies relating the lung microbiome with BPD development [26]. Infants who would later develop BPD had a different concentration of *Proteobacteria* and *Firmicutes*, a reduced number of *Lactobacilli* and an increased microbial turnover in the first weeks of life compared to those with no BPD. Since lactobacilli have established anti-inflammatory properties [109], such BPD predisposition could result in part from a shift to predominantly pro-inflammatory taxa. Lal et al. studied the airway microbiome at birth in 23 preterm infants (10 evolving to BPD and 13 not) and found that *Lactobacilli* were less abundant at birth in preterm neonates predisposed to BPD [110]. They then compared the microbiome examined at birth with the microbiome examined after diagnosis of BPD and found a subsequent increase in *Proteobacteria* phylum (with a predominance of *enterobacteriaceae*) and a decrease in the *Firmicutes* and *Fusobacteria* phyla. Lactobacilli were also less abundant in neonates born from mothers with chorioamnionitis. The composition of the fecal microbiota in a recent cohort of infants born at <29 weeks and diagnosed with BPD had significant differences in the relative abundance of *Klebsiella*, *Salmonella*, *Escherichia/Shigella* and *Bifidobacterium* species associated with down-regulation of immune-related genes by transcriptome sequencing analysis [111]. Lal et al. described an alternative mechanism by investigating pulmonary metabolome and correlating it with the microbiome [112]. The authors compared bronchoalveolar fluid samples of 30 preterm neonates at birth with and without BPD, showing an increase in *Proteobacteria* and a reduction in *Lactobacilli* associated with a decrease in the ratio acetyl-CoA/propionyl-CoA carboxylase, indicating a reduced fatty acid β-oxidation pathway and an increase in airway inflammation in the BPD group. Targeting specific microbiomes that lead to the metabolomic changes contributing to BPD with specific antimicrobial therapies could represent a novel preventive or therapeutic approach. Nevertheless, attempts made to decrease *Ureaplasma urealyticum* colonization by antibiotic prophylaxis did not result in the reduction of BPD rates [113].

## 5. Host and Microbiome Genetic Factors

Early small fingerprinting-based twin studies showed a greater similarity of microbiota in monozygotic vs heterozygotic twins [114], suggesting an influence of the host’s genetic background, but these results were not confirmed in 16S rRNA sequencing studies. Animal studies comparing microbiome diversity in different mouse strains showed that environmental factors were stronger determinants than the genetic background. Quantitative trait locus (QTL) analysis identified immunity-related genes influencing relative abundance of certain microbiota species, such as IRAK3 (regulating the toll-like receptor pathway), LYZ1/2, IFNγ and IL22. Single-gene approaches have identified the role of immune genes (MEFV, MYD88, NOD2, defensin-encoding genes, RELMB, HLA genes, the IgA locus), and metabolic genes (APOA1-encoding apolipoprotein, LEP and LEPR-encoding leptin and its receptor) (for review, see [115]). Subsequent host and microbiota genome-wide association studies (GWAS) confirmed the importance of immune gene variants (HLA-DRA, TLR1) but yielded divergent results, supporting the need for more omics research in this field [116].

## 6. Mechanisms Involved in the Placenta-Gut-Lung Cross-Talk

### 6.1. Antenatal Reciprocal Lung-Placenta Signaling

There is a subtle balance between maternal tolerance of the fetal immune system and resistance towards pathogens susceptible to infect the amnion and lead to premature labor and various neonatal morbidities, in which the genital tract microbiome plays a key role [117]. The transplacental mechanisms through which maternal taxonomy translates into the offspring is still poorly understood. Microbial metabolites may play a role in this exchange (see below).

Exosomes have emerged as mediators of cell-to-cell, organ-to-organ and system-to-system cross talk. Arterial cord blood obtained from neonates born at term after spontaneous labor compared to term and preterm cesarean-section-born infants, was shown to contain high levels of exosome-embedded complement component 4B-binding protein alpha chain (C4BPA). C’BPA originating from the fetal lung turns on several pro-labor placental genes such as corticotropin-releasing hormone (CRH), cyclo-oxygenase 2 (COX-2) and pro-inflammatory cytokines (TNFα, IL1, IL6, IL8) through CD40 interaction and NFκB activation [118]. A similar fetal–maternal interaction was demonstrated in mice by the induction of parturition through lung-derived steroid receptor coactivators 1 and 2 (SRC-1, -2) endogenous steroid pathway activation [119]. Preterm infant’s early dysbiosis, characterized by the predominance of gram-negative, potentially pathogenic *gammaproteobacteria*, is strongly associated with antenatal exogenous steroids exposure as well as inflammation [120]. Taken together, these studies suggest that the fetal/neonatal lung has the capacity of influencing the establishment and development of local and remote microbiomes, a process in which exosomes may play a key role.

### 6.2. Postnatal Immunity and Microbiomes

Respiratory and digestive mucosal surfaces are part of an integrated network of tissues, cells and effectors that constitute a global immunological organ in which stimulation of one compartment can lead to changes in distal areas. Oral, intestinal, airway and genital mucosae are interconnected through circulating lymphocytes [121]. The exposure to specific antigens, sensed by local dendritic cells, shape future immune response such as Th1 vs. Th2, with a longstanding impact on the health and diseases of the subject [122], a process in which the nature and composition of local microbiomes play a central role. As microbiota in each compartment have an influence on immune system development, the immune system plays a role in shaping and maintaining microbial communities, as demonstrated in germ-free zebrafish and mice [123]. A bilateral, immune-mediated gut-lung interaction shaping the respective microbiomes and influencing the health/disease balance in each organ is therefore plausible, but remains to be demonstrated in human neonates [124].

Bacterial metabolites and lung immune homeostasis: Gut microbiota generate metabolites during food assimilation that have direct effects on lung immune homeostasis (for review, see [125]). Among these, short chain fatty acids (SCFAs), in particular acetate, propionate and butyrate, reach the systemic circulation from the intestinal lymphatic system and modulate the lung immune balance. SCFAs promote regulatory T-cell generation and function through free fatty acid receptor 2 and 3 (FFAR 2/3) binding and histone deacetylase (HDAc) inhibition [126]. Mice with vancomycin-induced intestinal dysbiosis showing exacerbated Th-2 responses leading to allergic lung inflammation have been rescued by dietary SCFAs through attenuation of dendritic cell activation via programmed death ligand 1 (PD-L1) and decreased Immunoglobulin E (IgE) and interleukin 4 production [127]. Tryptophan metabolites generated by the intestinal microbiome, especially the lactobacillus genus, are natural ligands for the aryl hydrocarbon receptor (AhR), which in turn promotes regulatory T-cell development via interleukin 22 production [86]. In a human RCT of adults with emphysema, azithromycin treatment decreased intestinal microbiome diversity leading to increased bacterial metabolites such as glycolic acid, indol-3-acetate and linoleic acid, and reduced chemokine (C-X-C) ligand 1 (CXCL1), TNF)α, interleukin 13 and IL-12p40 in bronchoalveolar lavage fluid [128]. The role of these recently identified bioactive bacterial metabolites in neonatal lung disease are still to be unveiled.

### 6.3. Impact of Antibiotic Exposure

The wide exposure of infants to broad-spectrum antibiotics in the perinatal and neonatal periods has profound effects on microbiota. In a mouse BPD model, perinatal maternal antibiotic exposure increased pulmonary fibrosis, vascular remodeling, alveolar inflammation and mortality in offspring through disruption of commensal intestinal colonization, demonstrating a key role of the gut-lung axis [129]. Antenatal exposure to antibiotics can also influence a mother’s gut microbiota [130]. In a similar way, maternal chorioamnionitis can negatively influence neonatal gut microbiota and outcome. Puri at al. studied the fecal microbiome of 106 preterm infants ≤28 weeks [131]. Neonates born to mothers with chorioamnionitis had a higher abundance of *Bacterioides* and *Fusobacteria* in fecal samples, and were at higher risk of sepsis or death. Recently, a large meta-analysis showed that BPD, not RDS, is strongly associated with maternal chorioamnionitis [132]. Considering that women with chorioamnionitis almost invariably receive broad-spectrum antibiotics, the majority of their fetuses are exposed. Therefore, we can speculate that not only inflammation but also dysbiosis contributes to the higher risk of BPD associated with chorioamnionitis. The Canadian Healthy Infant Longitudinal Development CHILD study, a prospective cohort study of Canadian infants followed up over a period of two years, found that that an infant’s gut microbiota is persistently altered after intrapartum antibiotic prophylaxis exposure [133]. Does the dysbiosis induced postnatally by broad-spectrum antibiotics have an impact on the course of late-onset infectious disease? A retrospective paired cohort study comparing clinical courses in preterm infants with NEC suggests it’s not the case, showing no difference between those receiving probiotics vs those not [19]. Antibiotic-induced alterations of physiological gut microflora have been shown to last into adulthood [134]. A recent RCT on peripartum maternal antibiotics prophylaxis, demonstrating that azithromycin in addition to the standard cephalosporin regimen for women undergoing non-elective cesarean section decreased the risk of post-operative wound infections and other infectious complications, led to a broader fetal exposure to macrolides in the USA, but failed to address short-term pulmonary effects and long-term respiratory outcomes in newborns [135]. Some studies suggest that macrolides have beneficial short-term effects on lung disease of prematurity. Given the prevalence of *ureaplasma* species in chorioamnionitis and its suspected role in BPD, adding azithromycin to the standard antibiotic regimen in women with PPROM <28 weeks of gestation improves perinatal respiratory outcomes and the BPD risk [136]. However, the lasting impact of the microbiome is still poorly understood, and potentially significant. In a large Finnish cohort study, penicillin in early life had only a transient effect on intestinal microbiota, whereas macrolides correlated with substantial, long-standing shifts from normally dominant Gram-positive phyla to Gram-negative species [137]. This alteration was associated with asthma and obesity in children aged two to seven years. Therefore, the benefits and risks of perinatal broad-spectrum antibiotics, including macrolides, should be carefully investigated and weighed in future strategies.

The use of postnatal antibiotics reportedly modifies the existing flora, and reduced microbiome diversity is associated to BPD predisposition. In fact, antibiotic exposure in the first two weeks of life is associated with increased BPD rate and severity [138]. Broad-spectrum antibiotics increase the risk of multi-resistant bacterial infection, which in turn is associated with a twofold increase in BPD [139]. Furthermore, preterm infants with negative cultures exposed to more than five days of antibiotics have shown an increased BPD rate [140,141]. The Surveillance and Correction Of Unnecessary antibiotic Therapy SCOUT study, a multicenter observational study on antibiotic use in NICUs, showed that the majority of antibiotics are administered empirically and without clear evidence of infection [142].

Beyond antibiotics, other medications such as postnatal steroids influence the taxonomy and longitudinal development of microbiomes after birth. Grier et al. showed that steroids used for BPD prevention or treatment significantly altered the abundance of *bifidobacterium*, a beneficial intestinal genus, in a cohort of preterm infants followed longitudinally over the neonatal period [143]. The same authors showed that the lung and gut microbioma have distinct compositions, but develop in a tightly interdependent manner [144].

### 6.4. Acting on the Airway Microbiome

Conversely, would restoring a beneficial airway microbiome be a valid strategy for BPD prevention and treatment? Since the major source of microorganisms is the intestine, it is intuitive that the intestinal microbiome should translate into the lung microbiome, a hypothesis supported by some animal data [145]. Probiotics are live micro-organisms that, when administered in adequate amounts, confer a health benefit to the host. They exert a beneficial effect mostly by decreasing colonization and invasion by pathogenic organisms and by modifying the host immunity. Probiotic supplementation has proven to be effective for NEC prevention [22]. Animal studies have shown that enteral administration of probiotics impacts the respiratory microbiome [146], but no human studies have examined the effect of probiotic supplementation on the development of BPD as a primary outcome. Villamor-Martinez et al. conducted a systematic meta-analysis of 15 randomized trials on probiotics for NEC prevention and analyzed whether BPD incidence was affected in these studies [23]. While the meta-analysis confirmed a significant reduction of NEC (risk ratio (RR) 0.52, 95% confidence interval (CI) 0.33 to 0.81, *p* = 0.004), no significant effect on BPD could be demonstrated. In fact, every organ has a particular microbiome, and a probiotic beneficial for one system may not be for another. It is now accepted that differential bacterial strains of the same genus and species can have different effects on the host depending on the site of action. Therefore, every probiotic must be studied in order to act in a particular milieu [147]. The question whether intratracheal probiotic administration would improve respiratory outcomes in adults hospitalized in intensive care units is currently the object of a large RCT in Canada, the Prevention Of Severe Pneumonia and Endotracheal Colonization PROSPECT Trial [148]. If positive, this could represent an innovative approach for BPD prevention. In brief, modeling the lung microbiome is a potential alley for preventing BPD, but the modalities still have to be determined and properly tested.

## 7. Human Milk

Human milk plays a protective role towards the immature lung through various mechanisms including specific nutrients and factors, and through its influence on microbiota. A recent meta-analysis including 22 studies (17 cohort plus 5 RCTs) and 8661 infants [149] showed a trend towards a protective effect of human milk against BPD, calling for larger RCTs the determine a definite answer, which would undoubtedly raise ethical questions given its proven benefits in other diseases of prematurity such as necrotizing enterocolitis. In addition, differences exist between fresh maternal breast milk (MBM) and pasteurized human donor milk (DBM). In the largest meta-analysis to date based on 4984 infants, with 1416 BPD cases [150], there was a significant reduction of BPD incidence in exclusively MBM-fed infants compared with exclusive formula-fed (RR 0.74; 95% CI 0.57–0.96). Conversely, the same authors showed a trend but failed to formally demonstrate the same benefit comparing exclusive DBM and formula for BPD prevention (RR 0.89; CI 0.60–1.32), but suggested a protective effect of MBM vs. DBM (RR 0.77; 95% CI 0.62–0.96), even though the quality of evidence is low [18].

Many factors contribute to direct protective effects of MBM against prematurity co-morbidities, including specific macronutrients such as whey proteins, milk fat globules (MFG) and their specific lipids and membranes (MGFM), phosphoproteins and glycosylated proteins, oligosaccharides and many others [151]. In addition, antimicrobial factors such as lactoferrin, leucocytes, secretory IgA, complement factors, cytokines, lactoferrin and lysozyme play a fundamental immunomodulating role [152]. As recently discovered, human milk exosomes containing microRNA and other epigenetic factors interact with gene transcription and exert longstanding effects on gut homeostasis [153]. Pasteurization, which is considered a standard for DBM out of concerns for the transmission of infectious agents such as cytomegalovirus, Human Immunodeficiency Virus and Herpes Simplex Virus, inactivates many microcomponents and compromises its bactericidal and immunomodulating properties [152]. Alternative methods are currently under development, but have not reached prime time in the NICUs and DBM banks so far [154].

After birth, MBM promotes the colonization of the infant digestive tract and the development of the early microbiome [104] through various mechanisms. Bacteria are specific to the mother–infant dyad and the term of the pregnancy. In a 16S rRNA sequencing prospective study in healthy neonates, stable airway microbiomes characterized by predominant *moraxella* and *corynebacterium* species were associated with early breastfeeding and lower rates of infections in infancy [155]. Specific human milk oligosaccharides (HMOs), non-digestible prebiotics, act by promoting the development of *bifidobacteria* and *bacteroidetes*, which may have a beneficial effect on respiration and immunity. Indeed, HMO addition in formula provides protection against bronchiolitis in term infants [20]. However, HMO composition and concentration are significantly different in preterm compared to term breastmilk [156], and the effects of HMO on gut maturation and prevention of necrotizing enterocolitis appear to be age-dependent and uncertain in premature infants [21]. Up to now, evidence has been lacking indicating that early HMO supplementation has any beneficial effects in the preterm infant lung [21]. The influence of human milk nutrition on the airway microbiome in preterm infants and its potential effects in BPD prevention is still an open question.

## 8. Conclusions

Macro- and micronutrients have a profound influence on premature lung health and disease, raising the question of whether nutrition experts should be involved more systematically in neonatal intensive care teams. As bronchopulmonary dysplasia is a heterogeneous and multifactorial disease, single-bullet approaches have mostly failed to achieve significant progress. From this perspective, it is not surprising that a “simple” nutritional approach such as exclusive human milk feeding has shown an efficacy comparable to single-target drugs. The evidence of a neonatal lung microbiome playing a key role in RDS and BPD establishes new directions for prevention and therapeutic interventions, yet its origin, the dynamics of its establishment, its evolution and its interactions with the environment and its relationship with health maintenance and disease are still under intense investigation. The emerging concept of a placenta-gut-lung triangle should govern future research efforts, with the goal of preventing and potentially reversing the course of chronic lung disease of prematurity and preventing its lifelong consequences.

## Figures and Tables

**Figure 1 nutrients-12-00469-f001:**
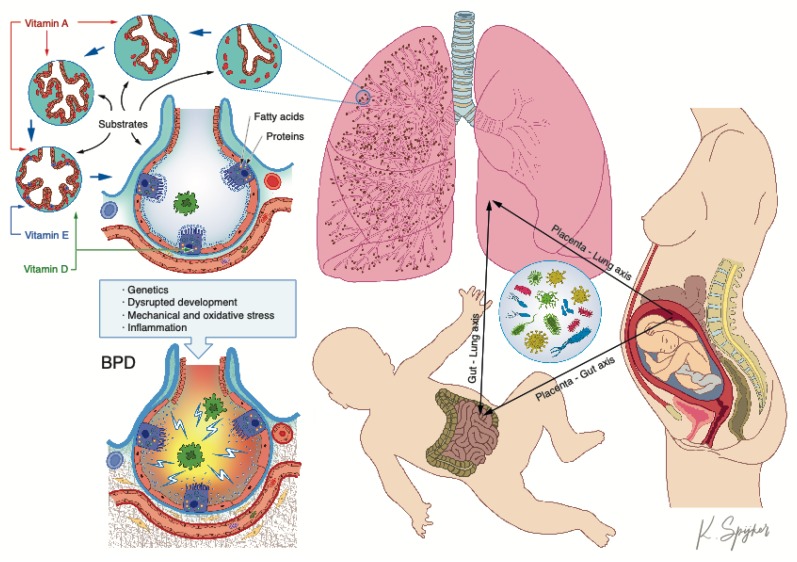
The placenta-gut-lung triangle. The figure illustrates the potential sources of the lung microbiome and the role of substrate supplies and specific nutrients in alveolar homeostasis, and their role in the pathway towards chronic lung disease.

**Table 1 nutrients-12-00469-t001:** Evidence-based recommendations on selected nutrients for premature lung disease prevention.

Nutrient		Mechanisms of Action	Effect on Lung Disease	RCT/Meta-Analyses	Recommendations
**Vitamins**	Vitamin A	Organogenesis, differentiation, lung development and growth, alveolar maturation	Low levels associated with decreased alveolarization and susceptibility to injury [2]	Antenatal supplementation in VitA deficit 7000 UI PO (RCT, [3])	Improved lung function—decreased morbidity **R (high)**
				Postnatal supplementation 5-10,000 UI IM (META–4 studies) [4]	Decreased BPD at 36 weeks **R (moderate)**
	Vitamin D	Pre/postnatal type II cell maturation, surfactant, immune system	Low levels associated with BPD, asthma, RTI	Postnatal supplementation 400–800 UI PO (2 RCT, [5,6])	No change in RDS and BPD NR (low) Decreased wheezing **R (moderate)**
	Vitamin E	Early lung development, antioxidant	Low levels associated with BPD	Postnatal supplementation 20–150 mg/kg IM/IV/PO (Cochrane [7]–26 studies)	No effect on BPD **NR (low)**
**Lipids**	LCPUFA	Organogenesis Modulation of inflammation	DHA/AA deficit increases BPD [8]	Postnatal ω-3 LCPUFA supplementation (META 26 14 RCT [9])	No effect on BPD **NR (high)**
			LCPUFA-rich emulsions increase DHA and EPA but decrease AA levels [10]	SMOF vs. Clinoleic-RCT-2 studies	Equivocal: BPD prevention [11] vs. no effect [12] **Unknown (low)**
				SMOF vs. Intralipid (Cochrane–15 studies [13])	Trend to BPD prevention **R (low)**
**Glucides**	Inositol	Surfactant synthesis	Improves early mortality and death/BPD [14]	Supplementation (80 mg/kg/day) (Cochrane–6 studies [15])	No effect on BPD **NR (high)**
**Peptides**	Lactoferrin	Bactericidal, innate immunity	Prevents NEC and LOS; unclear on BPD [16]	Supplementation 150 mg/kg 24–33 weeks (RCT [17]	No effect on BPD **NR (high)**
**Human milk**	Various	Various	MBM vs. formula decreased BPD; DBM vs. formula: trend to decreased BPD [18]	Combined exclusive vs. partial; MBM vs. DBM META 22 studies including 5 RCT [19]	Trend to BPD prevention (OR 0.73–1.03) **R (moderate)**
**Microbiome**	HMO (prebiotics)	Beneficial microbiota, gut maturation, immune system	HMO formula supplementation decreases bronchiolitis in term [20]; limited effects on immature organs [21]	No data on premature lung diseases	n/a
	Probiotics	Immune system, gut permeability, bacterial metabolites	Probiotics decrease NEC [22]	Supplementation for NEC as primary outcome META 15 studies [23]	No effect on BPD **NR (moderate)**

Abbreviations: LCPUFA: long chain polyinsaturatyed fatty acid; HMO: human milk oligosaccharide; BPD: bronchopulmonary dysplasia; RTI: respiratory tract infection; DHA: docosahexenoic acid; AA: arachidonic acid; NEC: necrotizing enterocolitis; LOS: late-onset sepsis; META: meta-analysis; MBM: maternal breast milk; DBM: donor breast milk; RCT: randomized controlled trial; RDS: respiratory distress syndrome; R: recommended; NR not recommended; high, moderate, low: level of recommendation.

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
