# Peer review of "Nutrients and Microbiota in Lung Diseases of Prematurity: The Placenta-Gut-Lung Triangle"

_nutrients, 2020, doi:10.3390/nu12020469_

Round 1

Reviewer 1 Report

We thank authors provide detail informations about our questions. 

Reviewer 2 Report

The authors provided an excellent revision that is approporiate for publication.

This manuscript is a resubmission of an earlier submission. The following is a list of the peer review reports and author responses from that submission.

Round 1

Reviewer 1 Report

The authors present a manuscript reviewing nutrients and microbiota in lung diseases of prematurity, especially focused on the “Placenta-Gut-Lung Triangle”. Overall the manuscript is well written and conveys important information that contributes to the field. Some edits are suggested to better clarify results.

The gut microbial community possesses enzymatic machinery for assimilating a variety of dietary nutrients leading to release of metabolites having multiple functions in the host. It is important to understand the influence of gut microbiota and metabolites produced by them on the functioning of various organs within the body. The gut microbiota affects other organs either by aspiration or produces metabolites that bring about changes when they reach other organs. However, in this article, the authors did not report the pathophysiology in this field. We recommended that a figure summarized the result. The influence of various types of dietary components on gut microbiome and associated physiological changes is important, especially focused on the “Placenta-Gut-Lung Triangle”. We recommended to list a table summarized the result. Author discussed several diseases in article. However, the suggestions were not clearly mentioned. A summarized table was suggested to include all recommendation. The genetic effect in “Placenta-Gut-Lung Triangle” is very important. It is important to mention in this article. The role of immune cells from placenta, gut to lung in mucosal immune system is an interesting issue and needed to discuss in this article The authors discussed several important nutrients, including Carbohydrates, Lipids, and Micronutrients. The recommendation of intake amount for neonatal mature, or diseases, was important for physicians and readers The relationship of lung disease mechanisms and nutrients is well investigated. We recommend author provided a figure to summary. Impact of antibiotic exposure in perinatal and neonatal periods is an important issue. In some condition, the administration of antibiotic for infection controlled is necessary. We suggested authors to provide the pros and cons in infectious gastroenterolitis and explained the mechanism of microbiota changed in gut-lung axis.

Reviewer 2 Report

The manuscript of Fiammetta Piersigilli et al entitled 'Nutrients and Microbiota in Lung Diseases of Prematurity: the placenta-Gut-Lung triangle' describes a very interesting review on the existing supporting evidence of the relationship between nutrients, microbiota, intra-uterine/placenta and post natal intestinal system in relationship to the development of BPD.

The manuscript is highly interesting en very well written. The authors succeeded to provide the available existing evidence, associations and data from adults on this topic that were translated to the idea of the triangle.

I am convinced about the potential effect of the placenta/microbiota in relation to the gut and how these potentially affect the lung of the preterm infant. I am not fully convinced, based on the provided data, on the effect of the lung on the gut and the placenta.... In other words should the arrows be in both directions? Maybe the authors can add a bit on the ideas

Next to that I was wondering if a part on the use of corticosteroids could be added. Current care includes the use of maternal corticosteroids in case of a risk of preterm birth and at the NICU corticosteroids are used to prevent and or treat BPD... How would this therapy fit within the concept of your paper?

Minor /typos: line 77_: 500-1000g instead of 100

line 489 HDM vs DBM... please chose one

line 529 placenta instead of uterus would be more consequent